

# Seasonal variation in apparent conductivity and soil salinity at two Narragansett Bay, RI salt marshes

Richard McKinney[1], Alana Hanson[1], Roxanne Johnson[1] and Michael Charpentier[2]

[1] Atlantic Coastal Environmental Sciences Division, US Environmental Protection Agency, Narragansett, RI, United States of America
[2] General Dynamics Information Technology, Narragansett, RI, United States of America

## ABSTRACT

Measurement of the apparent conductivity of salt marsh sediments using electro-magnetic induction (EMI) is a rapid alternative to traditional methods of salinity determination that can be used to map soil salinity across a marsh surface. Soil salinity measures can provide information about marsh processes, since salinity is important in determining the structure and function of tidally influenced marsh communities. While EMI has been shown to accurately reflect salinity to a specified depth, more information is needed on the potential for spatial and temporal variability in apparent conductivity measures that may impact the interpretation of salinity data. In this study we mapped soil salinity at two salt marshes in the Narragansett Bay, RI estuary monthly over the course of several years to examine spatial and temporal trends in marsh salinity. Mean monthly calculated salinity was 25.8 ± 5.5 ppt at Narrow River marsh (NAR), located near the mouth of the Bay, and 17.7 ± 5.3 ppt at Passeonkquis marsh (PAS) located in the upper Bay. Salinity varied seasonally with both marshes, showing the lowest values (16.3 and 8.3 ppt, respectively) in April and highest values (35.4 and 26.2 ppt, respectively) in August. Contour plots of calculated salinities showed that while the mean whole-marsh calculated salinity at both sites changed over time, within-marsh patterns of higher versus lower salinity were maintained at NAR but changed over time at PAS. Calculated salinity was significantly negatively correlated with elevation at NAR during a sub-set of 12 sample events, but not at PAS. Best-supported linear regression models for both sites included one-month and 6-month cumulative rainfall, and tide state as potential factors driving observed changes in calculated salinity. Mapping apparent conductivity of salt marsh sediments may be useful both identifying within-marsh micro-habitats, and documenting marsh-wide changes in salinity over time.

# INTRODUCTION

Salt marshes are productive ecosystems that by nature of their position in the landscape are subject to many natural and anthropogenic stressors. In the Northeast US there is concern about the impact of accelerated sea level rise on salt marsh hydrology (e.g., *Watson et al., 2017*), and how changes in marsh flooding might impact vegetation community structure

Corresponding author
Richard McKinney,
Mckinney.Rick@epa.gov

(*Smith et al., 2017*). Changes in vegetation communities may impact ecosystem services provided by salt marshes, and hence may have implications for their conservation and role in coastal ecosystems. For example, plant community structure can influence belowground biomass accumulation, which in northeastern US salt marshes is an important mechanism for marsh accretion that can mitigate the effects of sea-level rise (*Bricker-Urso et al., 1989*; *Turner, Swenson & Milan, 2000*). Alteration of vegetation community structure may also impact the provision of other ecosystem services such as nutrient storage, habitat availability for fauna, and fisheries production (*Kelleway et al., 2017*).

Tidal inundation is an important determinant of salt marsh vegetation community structure, realized in part through the species-specific differences in physiological responses of plants to salinity. As sea level rises the extent of tidal inundation will increase, potentially altering the distribution of plant species across a marsh. Since increased inundation will alter soil porewater salinity, and the primary route of water uptake in salt marsh plants is through porewater (e.g., *Al Hassan et al., 2017*), measurement of soil porewater salinity could provide insight into potential vegetation community changes resulting from sea-level rise (*Silvestri & Marani, 2004*). However, few studies have examined whole-marsh porewater salinity, in part because of the labor-intensive sampling required and the difficulty in consistently obtaining porewater samples at depth. An alternative is to estimate salt marsh porewater salinity by measuring the apparent conductivity ($EC_a$) of salt marsh sediments using electromagnetic induction, which can generate sufficient data over the course of several hours to map soil salinity across a marsh surface. This approach provides estimates of soil salinity even in areas where the saturated zone is deep, or where there are clay or fine sediment layers with low hydraulic conductivity rendering porewater difficult to sample.

Measurement of $EC_a$ in soils has been used since the mid-20th century to aid in mineral and petroleum exploration and extraction, and over the past 40 years to characterize the salinity of agricultural soils (*DeJong et al., 1979*). More recently the emergence of portable instrumentation capable of rapid field measurements has allowed for its use in the estimation of other soil parameters (*Robinson et al., 2004*). In simplest terms, at a given temperature $EC_a$ is primarily influenced by four characteristics: soil composition, i.e, mineral or clay content; bulk density; moisture content; and ion concentrations, which can be representative of soil salinity (*Corwin & Lesch, 2005*). Each of these characteristics affects the bulk conductivity of soils, which in turn influences the extent to which an induced electromagnetic field can be generated through the soil. $EC_a$ is determined by measuring this induced electromagnetic field, which in turn reflects the average conductivity, influenced by all soil characteristics, over a volume of soil (*Doolittle, Petersen & Wheeler, 2001*). Differences in instrument response can be experimentally calibrated to changes in a selected soil characteristic, allowing, under the assumption that all other characteristics are constant, for a proxy measure of changes in that characteristic in the soil.

Application of $EC_a$ measures in salt marshes to map soil porewater salinity was first explored in the early 2000s (*Paine et al., 2004*) but later developed by *Moore et al. (2011)*. The approach uses an electromagnetic induction (EMI) instrument to measure $EC_a$ at a series of sample points across a marsh surface. At a subset of sample points,

$EC_a$ is calibrated with soil porewater salinity, measured using a sipper technique (*Portnoy & Valiela, 1997*). The resulting calibration curve is then used to calculate salinity based solely on $EC_a$, which can then be mapped in a GIS to develop contours of salinity values across the marsh surface. This technique has been used to examine the relationship between plant species distribution and soil salinity during the growing season, but to our knowledge no earlier studies have looked at inter-annual changes in soil salinity patterns. In this study, we measured $EC_a$ across two southern New England salt marshes along an estuarine salinity gradient over a period of 2 years to investigate intra-marsh variability in soil salinity, as well as potential drivers of seasonal changes in mean salinity observed at each marsh. The underlying assumption of this technique is that in uniformly saturated soils, such as those found in salt marshes, the contribution of soil moisture content to $EC_a$ will be constant, and that variability contributed by other soil characteristics is limited, such that changes in $EC_a$ values will accurately reflect changes in porewater salinity. To begin to evaluate the validity of this assumption, we also examined changes in the relationship of $EC_a$ and measured porewater salinity at our sites with respect to potentially confounding factors such as bulk density, percent moisture of the soil, and marsh elevation. Our results will provide information about the magnitude of seasonal salinity change observed at a marsh, as well as identify potential drivers of that change. Our study will also aid in evaluating $EC_a$ as a surrogate for porewater salinity, provide insight into potential factors influencing $EC_a$ in salt marsh soils, and help identify environmental factors that could confound the relationship between $EC_a$ and salinity. This information may allow for more widespread application of the technique, for example to use in monitoring the trajectory of marsh degradation or recovery during salt marsh restoration efforts.

## MATERIALS & METHODS

### Site descriptions

The study area was two salt marshes sites located in the Narragansett Bay estuary, Rhode Island, USA (Fig. 1). The southern site (NAR) was near the mouth of the Pettaquamscutt sub-estuary (41°26′49.6″N, 71°26′58.0″W), and had a total area of 5.89 ha. The upland edge of the site was bordered by an equal proportion of private residences and forest habitat. The marsh surface consisted of low marsh habitat dominated by short form *Spartina alterniflora*, and high marsh habitat dominated by *Spartina patens*, *Distichlis spicata*, and *Juncus gerardii*. The high marsh—upland border consisted primarily of *Iva frutescens*, and small patches of *Typha spp.* and *Schoenoplectus spp.* The northern site (PAS) was within the Passeonkquis Cove sub-estuary (41°44′52.8″N, 71°23′5.2″W), and had a total area of 2.35 ha. The upland edge of the site was bordered by an approximately 100m-wide patch of trees and dense understory vegetation, transitioning to dense residential land use. The marsh surface consisted of low marsh habitat dominated by tall form *Spartina alterniflora*, and high marsh habitat dominated by *Spartina patens* and *Distichlis spicata*. The high marsh—upland border consisted primarily of *Iva frutescens*, with a 0.68 ha patch of Typha *spp*. at the northern edge of the border.

# PeerJ

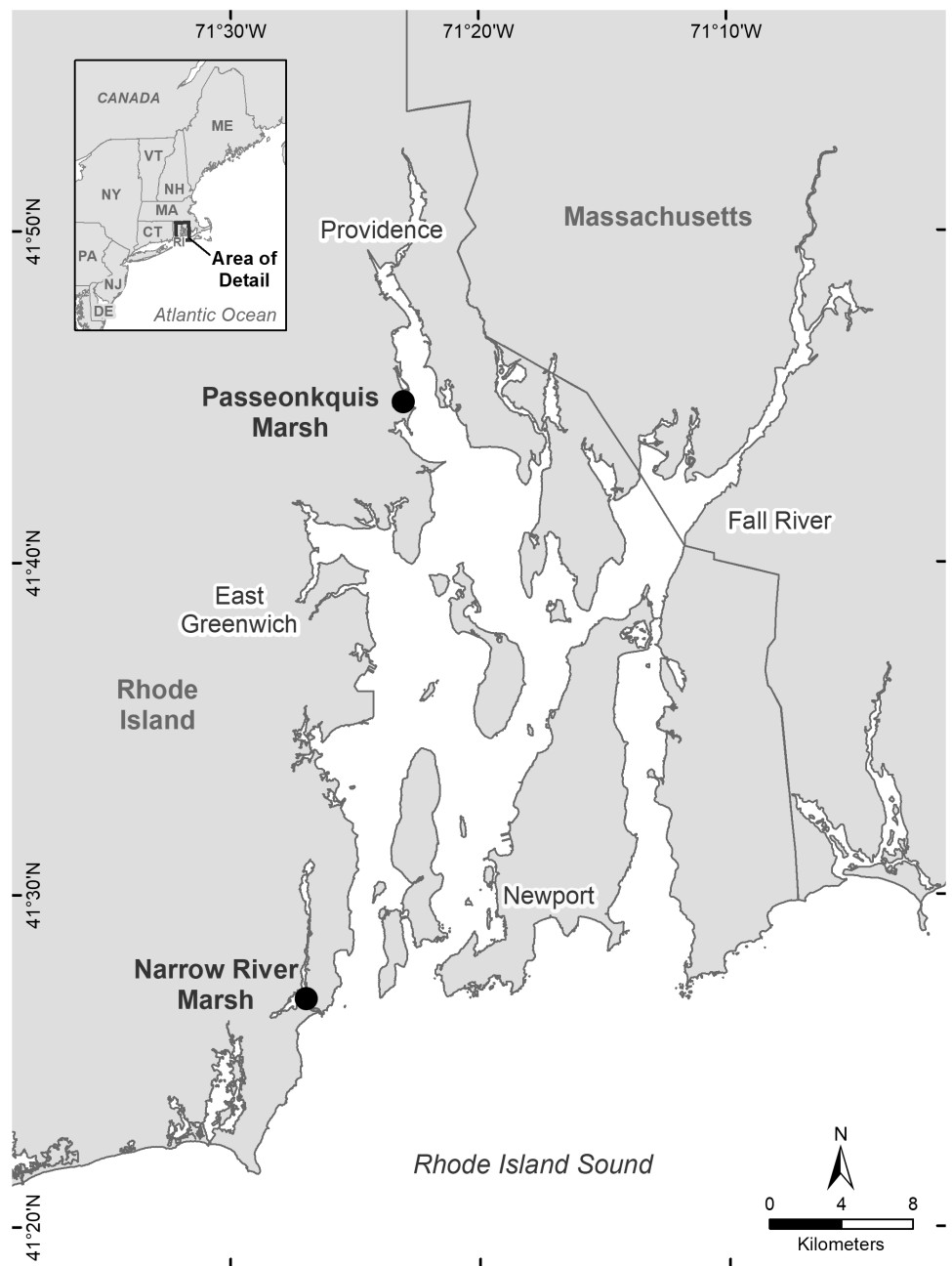

**Figure 1** Location of the two salt marsh study sites Narrow River marsh (NAR) and Passeonkquis marsh (PAS) in the Narragansett Bay estuary, Rhode Island, USA.

## Field measurements

A Geonics Model EM38-MK 2 Conductivity Meter (Geonics Ltd, Mississauga, Ontario, Canada) was used in horizontal mode, held 50 cm over the marsh surface, to record $EC_a$ readings. The readings were the result of an induced current generated by the instrument through a maximum penetration depth of approximately 1.0 m of soil at randomly

distributed sample points across each marsh surface (*Geonics Limited, 2014*). $EC_a$ values in milliSeimans meter$^{-1}$ (mS m$^{-1}$) along with the latitude and longitude of the sample point and vegetation characteristics were entered into an ArcGIS shapefile using ArcPad software (ESRI, Redlands, CA) on a Trimble Nomad hand-held field computer (Trimble Navigation Ltd., Sunnyvale, CA USA). Samples were taken approximately every 30 days beginning October, 2015 through October 2017 ($n = 24$ sample events). Both sites were surveyed on the same day at approximate 10:00 am (NAR site) and 12:00 pm (PAS site). The surveys consisted of a random transect pattern walked across the marsh surface, with $EC_a$ values, vegetation characteristics, and sample point position recorded approximately every 5 m. Porewater salinity measures were taken at a randomly selected sub-set of sample points (mean frequency of 29.6% of the points across both marshes) using a sipper consisting of a 0.5 m long piece of 1.0 mm diameter serrated metal tubing inserted in the soil to a depth of 0.25 m. Once inserted, approximately 25 ml of porewater was withdrawn and its salinity measured using a refractometer. Porewater salinity readings, when taken, were also stored in the ArcGIS shapefile.

Following field sampling, shapefiles were transferred to a GIS where contour maps of calculated salinity across each marsh surface were created using the ArcGIS version 10.3 Spatial Data Analyst, inverse distance-weighted interpolation function (ESRI, Redlands, CA). $EC_a$ data were first converted to calculated salinity values using marsh and survey-specific calibration curves constructed from a least-squares regression of $EC_a$ values and measured porewater salinities. Calculated salinity values were then used in the ArcGIS software inverse distance-weighted interpolation function to create marsh-specific contour maps for each sample event.

Elevation values were collected using an RTK GPS Global Navigation Satellite System (GNSS) receiver (Trimble Navigation Limited, Dayton, Ohio) at approximately 100 locations per marsh. Each sample location was selected at approximately 5 m intervals along randomly-placed transects across the marsh surface. Elevations were referenced to nearby benchmarks, and the WGS84 ellipsoid model was used to determine vertical and horizontal position. The National Geodetic Survey Geoid 12A (CONUS) model was used to calculate elevations from orthometric heights (North American Vertical Datum of 1988 [NAVD88]), and all points were projected to North American Datum of 1983 (NAD83) Universal Transverse Mercator zone 19. Digital elevation models (DEMs) were created from survey points using the inverse distance weighting function in ArcGIS software. Elevation values corresponding to sample point locations were interpolated from the DEMs for 12 sample events corresponding to maxima and minima values of mean whole-marsh calculated salinity. Three sample events were chosen to bracket each of two occurrences of maxima and minima over the course of the study. Interpolated elevations ranged from 0.24 to 0.76 m above mean sea level (MSL) for NAR, and from 0.49 to 1.04 ft above MSL for PAS. We estimated bulk density and moisture content of soil by collecting 6 soil cores of 25 cm depth along a randomly-placed transect from the upland to seaward edge at each site. Two cores were collected at the mid-point of the high and low marsh zones as determined by dominant plant species, and at the mid-point of the transect (mid marsh). Each core was sectioned in 5 cm increments and a soil subsample from each depth was weighed, dried,

and then re-weighed to determine bulk density and percent moisture. Permission for this non-invasive field study was provided by RI Department of Environmental Management, under collection permits #2015-31-F–2018-31-F.

Total rainfall was obtained from the NOAA National Centers for Environmental Information, Climate Data Online website (https://www.ncdc.noaa.gov/cdo-web/) for the stations Kingston, RI (41°29′25.1″N, 71°32′34.8″W), located approximately 9 km northwest of NAR, and Providence, RI (41°50′33.7″N, 71°23′6.7″W), located approximately 10.5 km north of PAS. Daily rainfall amounts were aggregated into cumulative amounts over 24 h, 36 h, 1 month, 3 month, and 6 month periods prior to each sample event. Using Spearman Rank Correlation analysis we found that 24 h and 36 h values were significantly correlated ($r^2 = 0.88$, $p = 0.001$), as were 1 month and 3 month cumulative values ($r^2 = 0.45$, $p = 0.001$). We therefore included only 24 h, 1 month, and 6 month cumulative rainfall in linear regression models to examine the effect of cumulative rainfall and tide height on mean calculated salinity. Tide heights were obtained using online tide charts containing the time of low and high tides and corresponding tide heights relative to mean low water (NOAA Tides and Currents, https://tidesandcurrents.noaa.gov/). We used data from sites at Narragansett Pier, RI (41°25′56.0″N, 71°27′25.2″W) located approximately 2 km south of NAR, and Pawtuxet Cove, RI (41°44′53.6″N, 71°23′0.6″W) located approximately 1.3 km north of PAS. Tide height was extrapolated at time of sampling from predicted tide ranges and expressed as a proportion of the maximum tide height for the tide cycle during which the sample occurred.

## Data analysis

We examined temporal variability in calculated salinity for each marsh by plotting mean calculated salinity versus sample date. Calculated salinities were derived from $EC_a$ data that were converted to calculated salinity values using marsh and survey-specific calibration curves constructed from a least-squares regression of $EC_a$ values and measured porewater salinities at points where sipper measurements were taken. The slopes of the calibration curves ranged from 0.018 to 0.081 (mean $0.044 \pm 0.016$) at NAR, and from $-0.099$ to 0.144 (mean $0.054 \pm 0.016$) at PAS. Coefficients of determination ranged from 0.12 to 0.92 (mean $0.49 \pm 0.19$) at NAR, and from 0.01 to 0.84 (mean $0.40 \pm 0.22$) at PAS.

The effect of cumulative rainfall and tide height on mean calculated salinity in the marsh was examined by constructing a series of linear regression models and evaluating the models using small sample Akaike Information Criteria ($AIC_c$), which accounts for biases that might arise from relatively small sample size (*Burnham & Anderson, 2002*). Candidate linear regression models ($n = 15$) were ranked by computing $AIC_c$ differences or Akaike weights as $\Delta AIC_c = AIC_{ci} - AIC_{cmin}$ (*Burnham & Anderson, 2002*, pp. 70–72). We then selected models best supported by the data as having $\Delta AIC_c$ values between 0.00 and 2.00 (*Burnham & Anderson, 2002*, pp. 75–77), and calculated the relative importance ($w + (j)$) of each parameter by summing the Akaike weights of all models that included this characteristic (*Burnham & Anderson, 2002*, pp. 167–169). Relative importance values provide a means to incorporate selection uncertainty in the evaluation of a set of parameters, and larger values of $w + (j)$ indicate whether a parameter may be a better predictor variable

(*Burnham & Anderson, 2002*). Statistical analyses were performed with SAS for Windows ver. 9.41 (SAS Institute, Inc., Cary, NC, USA).

We examined intra-marsh spatial and temporal variability in calculated salinity by plotting calculated salinity versus elevation at the sample points. We used least-squares regression of calculated salinity and corresponding elevation values obtained using the DEM for a given marsh and sample event for a sub-set of 12 sampling events chosen to correspond with maxima and minima in mean salinity values observed over time. We then compared regression statistics to trends in overall mean salinity for each marsh over time.

## RESULTS

For the NAR marsh, salinity was high at the seaward edge and low at the terrestrial border across the spring to fall growing season (Fig. 2). For PAS, contour plots show a more uniform distribution of salinity values across the marsh surface, particularly at calculated salinity minima (Fig. 3). During the October 2017 calculated salinity maximum, there was some evidence of a pattern of lower salinity towards the upland border (Fig. 3A, upper edge of the marsh in the plot), but that pattern was not evident during the other maximum or the minima.

The calibration coefficients for the least-squares regressions of porewater salinity versus conductivity for the 24 sample events ranged from 0.13 to 0.92 at the NAR site and 0.01–0.75 at PAS (Table 1). The coefficients, as well as error prediction parameters, were highly variable between events, without any consistent patterns or trends in the values at either site. Calculated salinities for each sampling event at NAR ranged from 16.3 to 35.4 ppt, with an overall mean for the entire study of 25.8 ± 5.5 ppt (Table 2). Calculated salinities at PAS ranged from 8.3 to 26.2 ppt, with an overall mean for the entire study of 17.7 ± 5.3 ppt (Table 2). While the overall mean calculated salinity showed a difference of 8.1 ppt between the two sites, for a given sample event the differences varied from 0.8 to 15.8 ppt. Mean calculated salinities for both sites showed maxima during the September 16, 2016 and October 27, 2017 sampling events (Fig. 4). Mean calculated salinities showed minima during the May 4, 2016 and June 15, 2017 sampling events for NAR, and the May 4, 2016 and July 21, 2017 sampling events for PAS (Fig. 4).

Linear regression models of whole-marsh calculated salinity versus environmental factors best supported by the data included one-month and 6-month cumulative rainfall and tide state for both sites (Table 3). At NAR, 6-month cumulative rainfall had the highest relative importance, about 1.7 times that of tide state and 3 times that of one-month cumulative rainfall (Table 4). At PAS, the factors 6-month and one-month cumulative rainfall had essentially equivalent relative importance, slightly greater than that of tidal height (Table 4).

Soil bulk density ranged from 0.16 to 0.34 g cm$^{-3}$ across the two sites (Table 5) and differed among marsh zones at PAS (ANOVA: $df = 2$, $F = 7.952$, $p = 0.006$; Tukey–Kramer test: low marsh differs significantly from mid and high marsh). Soil percent moisture ranged from 69.0 to 84.3% (Table 5), and, when averaged across the entire marsh, was greater at PAS (81.5 ± 3.6%) than at NAR (74.9 ± 11.5%; $t$-test: $df = 28$; $t = 2.105$; $p = 0.044$).

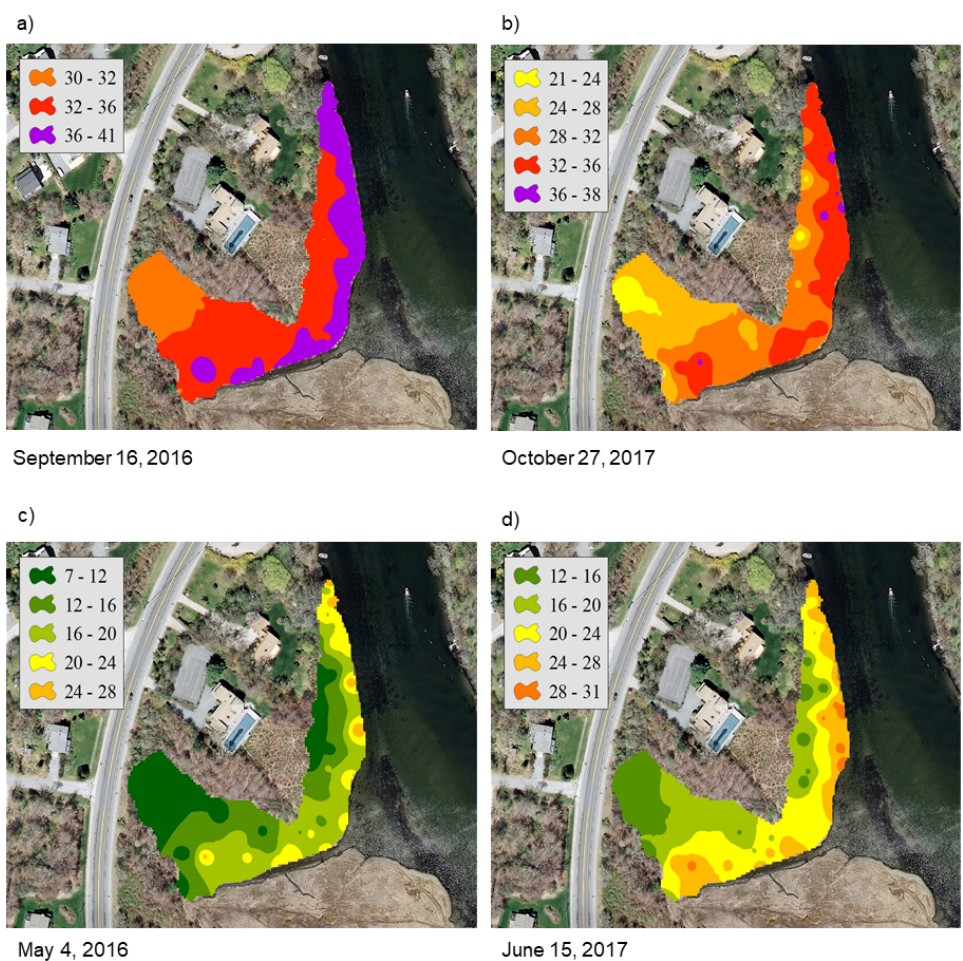

**Figure 2** Contour plots of calculated salinity across the marsh surface of NAR generated using inverse distance weighted interpolation corresponding to the (A) and (B) mean calculated salinity maxima, and (C) and (D) mean calculated salinity minima.

Percent moisture also differed among zones at PAS (ANOVA: $df = 2, F = 27.276, p < 0.001$; Tukey–Kramer test: low marsh differs significantly from mid and high marsh).

Calculated salinity was significantly negatively correlated with elevation at NAR during the 12 sample events on or around the salinity maxima and minima (Table 6). Slopes of the regression equations did not differ significantly between the maximum and minimum events. Elevation at NAR across all sample points during the 12 sample events ranged from 1.06 to 1.90 ft, with a mean of 1.54 ft. At the PAS site, calculated salinity was significantly negatively correlated with elevation for only the June and August 2017 maxima, and the October 2016 and September 2017 minima (Table 6). At PAS slopes of the regression equations also did not differ significantly between the maximum and minimum events. Elevation at PAS across all sample points during the 12 sample events ranged from 0.62 to 1.02 m, with a mean of 0.87 m.

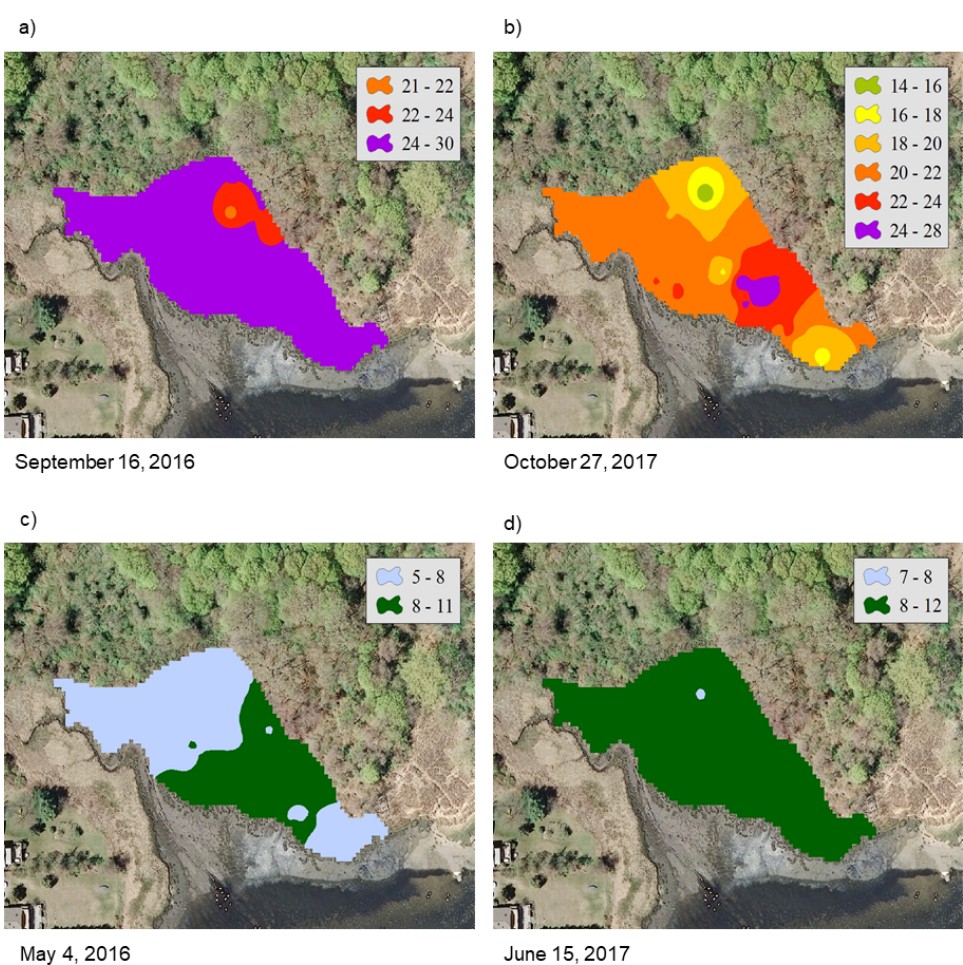

**Figure 3** Contour plots of calculated salinity across the marsh surface of PAS generated using inverse distance weighted interpolation corresponding to the (A) and (B) mean calculated salinity maxima, and (C) and (D) mean calculated salinity minima.

## DISCUSSION

Contours of calculated salinity showed both inter- and intra-marsh differences at our sites, and differences were variable over time throughout the study. At NAR, contours showed an expected pattern of soil salinity with higher values near the creek edge and lower towards the upland border. This pattern was generally maintained except during one period of high salinity incorporating the September 16, 2016 sampling event, when overall marsh salinity was at its highest. This sampling event followed a period of relatively severe drought in the region which occurred from early spring through the fall of 2016. During this drought, the impact of evapotranspiration on marsh hydrology may have been more pronounced and could have resulted in reduced groundwater recharge from uplands, and hence greater seawater influence. Several studies have modeled salt marsh groundwater dynamics and water table position by considering groundwater flow as a shallow, rigid aquifer in contact with a sinusoidally oscillating reservoir, and predicted the potential for greater seawater
**Table 1** Coefficients of calibration (root mean square error dependent means), sum of squared residuals, predicted residual sum of squares, and calibration coefficients from least squares regressions of apparent conductivity (EC$_a$) values and measured porewater salinities for 24 sample events at the (a) southern (NAR) and (b) northern (PAS) study sites.

**(a.)**

| NAR sample date | Coefficient of variation | Sum of squared residuals | Predicted residual sum of squares | Calibration coefficient ($r^2$) |
|---|---|---|---|---|
| 30 Oct 2015 | 29.842 | 79191 | 110199 | 0.78 |
| 4 Dec 2015 | 39.236 | 97394 | 122353 | 0.56 |
| 30 Dec 2015 | 52.270 | 180068 | 223680 | 0.46 |
| 29 Jan 2016 | 49.536 | 180847 | 249408 | 0.16 |
| 7 Mar 2016 | 27.860 | 40988 | 64366 | 0.66 |
| 6 Apr 2016 | 63.626 | 156358 | 235861 | 0.13 |
| 4 May 2016 | 30.083 | 94784 | 124701 | 0.25 |
| 1 June 2016 | 18.678 | 15643 | 22136 | 0.92 |
| 24 June 2016 | 29.803 | 121322 | 164145 | 0.63 |
| 4 Aug 2016 | 29.259 | 182849 | 242562 | 0.44 |
| 16 Sept 2016 | 28.051 | 199277 | 271598 | 0.27 |
| 7 Oct 2016 | 22.674 | 128346 | 164483 | 0.61 |
| 28 Oct 2016 | 31.011 | 245924 | 303635 | 0.42 |
| 1 Dec 2016 | 28.961 | 530237 | 612677 | 0.41 |
| 30 Dec 2016 | 49.937 | 311022 | 367100 | 0.37 |
| 29 Jan 2017 | 34.739 | 192455 | 241947 | 0.41 |
| 2 Mar 2017 | 40.657 | 182386 | 205666 | 0.66 |
| 29 Mar 2017 | 51.991 | 285369 | 354594 | 0.37 |
| 3 May 2017 | 34.691 | 73862 | 218634 | 0.36 |
| 15 June 2017 | 43.591 | 265452 | 320012 | 0.44 |
| 21 July 2017 | 17.821 | 63612 | 101593 | 0.77 |
| 16 Aug 2017 | 32.059 | 293982 | 349819 | 0.51 |
| 14 Sept 2017 | 26.958 | 175088 | 208907 | 0.67 |
| 27 Oct 2017 | 28.873 | 153907 | 185583 | 0.54 |

**(b.)**

| PAS sample date | Coefficient of variation | Sum of squared residuals | Predicted residual sum of squares | Calibration coefficient ($r^2$) |
|---|---|---|---|---|
| 30 Oct 2015 | 19.582 | 17318 | 26474 | 0.72 |
| 4 Dec 2015 | 15.152 | 5285 | 14160 | 0.12 |
| 30 Dec 2015 | 19.609 | 8638 | 17363 | 0.58 |
| 29 Jan 2016 | 22.107 | 11722 | 18318 | 0.31 |
| 7 Mar 2016 | 16.840 | 3413 | 6002 | 0.53 |
| 6 Apr 2016 | 14.834 | 2537 | 10192 | 0.34 |
| 4 May 2016 | 21.255 | 8471 | 15126 | 0.33 |
| 1 June 2016 | 21.049 | 8861 | 20611 | 0.13 |
| 24 June 2016 | 19.900 | 15849 | 49975 | 0.01 |

**Table 1** (*continued*)

**(b.)**

| PAS sample date | Coefficient of variation | Sum of squared residuals | Predicted residual sum of squares | Calibration coefficient ($r^2$) |
|---|---|---|---|---|
| 4 Aug 2016 | 8.183 | 4505 | 7272 | 0.65 |
| 16 Sept 2016 | 14.158 | 18307 | 29701 | 0.19 |
| 7 Oct 2016 | 12.061 | 12432 | 16553 | 0.42 |
| 28 Oct 2016 | 13.408 | 14472 | 21083 | 0.29 |
| 1 Dec 2016 | 15.753 | 12759 | 17545 | 0.28 |
| 30 Dec 2016 | 13.911 | 6212 | 8928 | 0.59 |
| 29 Jan 2017 | 20.937 | 14154 | 20493 | 0.20 |
| 2 Mar 2017 | 22.898 | 20047 | 31109 | 0.29 |
| 29 Mar 2017 | 18.886 | 12726 | 15743 | 0.52 |
| 3 May 2017 | 20.705 | 9135 | 13823 | 0.59 |
| 15 June 2017 | 16.959 | 11120 | 18111 | 0.75 |
| 21 July 2017 | 18.769 | 10313 | 21309 | 0.13 |
| 16 Aug 2017 | 15.431 | 14303 | 19796 | 0.16 |
| 14 Sept 2017 | 14.038 | 14023 | 22991 | 0.52 |
| 27 Oct 2017 | 13.628 | 10648 | 18538 | 0.59 |

inflow in the absence of groundwater inputs (*Montalto, Parlange & Steenhuis, 2007*; *Li & Jiao, 2003*). In northeast US salt marshes, seawater influence has been shown to diminish as distance from tidal creeks increases (*Hemmond & Fifield, 1982*), but during periods of extreme drought and lowered water table levels the effects of seawater inundation may be seen even in more interior portions of the marsh. However, globally many factors affect the characteristic salinity of tidal wetlands, and patterns that we observe locally may not be apparent depending on wetland type and location (*Mitsch & Gosselink, 2000*). For example, extensive freshwater inflow contributes to the characteristic salinities observed in Mississippi delta, USA wetlands, and marshes can also be influenced by seawater inflow can also exhibit uniform patterns of high soil salinity. Other examples of unique salinity patterns in tidal wetlands include observed wider ranges of salinity in Australian mangroves (*Boto & Wellington, 1984*), and higher overall salinities in Hudson Bay, Canada wetlands that are attributed to fossil salt deposits (*Price & Woo, 1988*).

At PAS, intra-marsh differences were not as distinct, and the marsh often showed homogeneous salinity patterns exemplified by the September 16, 2016 and July 21, 2017 sample events. This may have been a result of the marsh having a relatively small surface area, or of enhanced surface freshwater and groundwater inputs. Elevation increases rapidly in the upland area immediately bordering the marsh, and there is a small stream bordering the western portion. If the steep elevation serves to focus groundwater to the marsh, that along with the presence of the stream may result in lower salinity levels during times of the year when there is little evapotranspiration, and the effect may predominate over that of tidal inundation.

The NAR site is in the southern portion near the mouth of the Narragansett Bay estuary, and this probably accounts for its measured mean whole-marsh calculated salinity being

**Table 2 Mean whole-marsh conductivity (±SE), measured porewater salinity, and calculated salinity, and coefficients from least squares regressions used for calibration for 24 sample events at the (a) southern (NAR) and (b) northern (PAS) study sites.** Conductivity and calculated salinity were averaged across all sample points on the marsh surface, and measured porewater salinity was averaged across the sub-set of sample points where porewater was collected. Calibration coefficients for the corresponding calibration curves were constructed from a least squares regression of apparent conductivity ($EC_a$) values and measured porewater salinities.

**(a.)**

| NAR sample date | Mean conductivity (mS m$^{-1}$) | Mean measured porewater salinity (ppt) | Mean calculated salinity (ppt) | Calibration Coefficient ($r^2$) |
|---|---|---|---|---|
| 30 Oct 2015 | 317.7 ± 24.5 | 24.3 ± 4.0 | 23.4 ± 1.5 | 0.78 |
| 4 Dec 2015 | 238.4 ± 18.8 | 25.1 ± 2.9 | 26.2 ± 1.3 | 0.56 |
| 30 Dec 2015 | 249.4 ± 21.4 | 23.7 ± 3.1 | 24.5 ± 1.0 | 0.46 |
| 29 Jan 2016 | 222.7 ± 20.5 | 25.6 ± 2.4 | 24.7 ± 0.8 | 0.16 |
| 7 Mar 2016 | 213.6 ± 20.2 | 22.8 ± 3.8 | 18.2 ± 1.6 | 0.66 |
| 6 Apr 2016 | 191.6 ± 17.8 | 21.4 ± 2.3 | 21.5 ± 0.8 | 0.13 |
| 4 May 2016 | 193.6 ± 18.3 | 20.0 ± 2.5 | 16.3 ± 0.9 | 0.25 |
| 1 June 2016 | 261.5 ± 24.3 | 15.9 ± 3.8 | 17.7 ± 1.6 | 0.92 |
| 24 June 2016 | 305.7 ± 23.2 | 22.7 ± 2.3 | 20.3 ± 1.0 | 0.63 |
| 4 Aug 2016 | 402.6 ± 21.9 | 34.0 ± 2.0 | 33.9 ± 1.3 | 0.44 |
| 16 Sept 2016 | 389.0 ± 21.6 | 36.1 ± 1.3 | 35.4 ± 0.4 | 0.27 |
| 7 Oct 2016 | 388.1 ± 18.0 | 34.4 ± 1.3 | 34.6 ± 1.1 | 0.61 |
| 28 Oct 2016 | 386.7 ± 19.4 | 35.3 ± 1.4 | 35.2 ± 1.0 | 0.42 |
| 1 Dec 2016 | 491.5 ± 33.9 | 32.7 ± 1.4 | 31.2 ± 1.1 | 0.41 |
| 30 Dec 2016 | 263.1 ± 20.2 | 29.6 ± 2.3 | 28.8 ± 1.0 | 0.37 |
| 29 Jan 2017 | 269.6 ± 16.9 | 28.8 ± 2.7 | 27.1 ± 1.2 | 0.41 |
| 2 Mar 2017 | 252.2 ± 19.4 | 21.3 ± 2.4 | 22.6 ± 1.3 | 0.66 |
| 29 Mar 2017 | 253.0 ± 22.5 | 23.5 ± 2.1 | 23.1 ± 1.0 | 0.37 |
| 3 May 2017 | 245.0 ± 16.9 | 24.7 ± 1.9 | 23.0 ± 1.0 | 0.36 |
| 15 June 2017 | 295.7 ± 20.7 | 21.6 ± 2.1 | 21.3 ± 0.9 | 0.44 |
| 21 July 2017 | 347.8 ± 19.4 | 26.6 ± 2.1 | 24.9 ± 1.2 | 0.77 |
| 16 Aug 2017 | 353.9 ± 18.4 | 27.9 ± 1.9 | 26.7 ± 0.9 | 0.51 |
| 14 Sept 2017 | 381.3 ± 16.8 | 27.0 ± 2.1 | 27.6 ± 0.9 | 0.67 |
| 27 Oct 2017 | 339.4 ± 18.6 | 30.5 ± 1.5 | 30.2 ± 1.0 | 0.54 |

**(b.)**

| PAS sample date | Mean conductivity (mS m$^{-1}$) | Mean measured porewater salinity (ppt) | Mean calculated salinity (ppt) | Calibration coefficient ($r^2$) |
|---|---|---|---|---|
| 30 Oct 2015 | 249.7 ± 9.5 | 20.5 ± 2.0 | 21.3 ± 1.0 | 0.72 |
| 4 Dec 2015 | 207.7 ± 7.5 | 21.9 ± 1.4 | 22.4 ± 0.9 | 0.12 |
| 30 Dec 2015 | 193.8 ± 7.6 | 16.6 ± 2.8 | 16.7 ± 1.2 | 0.58 |
| 29 Jan 2016 | 176.7 ± 7.8 | 15.1 ± 1.3 | 14.7 ± 1.1 | 0.31 |

**Table 2** (*continued*)

**(b.)**

| PAS sample date | Mean conductivity (mS m⁻¹) | Mean measured porewater salinity (ppt) | Mean calculated salinity (ppt) | Calibration coefficient ($r^2$) |
|---|---|---|---|---|
| 7 Mar 2016 | 151.7 ± 6.4 | 11.9 ± 1.3 | 17.4 ± 0.7 | 0.53 |
| 6 Apr 2016 | 146.9 ± 6.7 | 10.9 ± 1.6 | 11.4 ± 1.0 | 0.34 |
| 4 May 2016 | 154.1 ± 6.7 | 8.7 ± 1.3 | 8.3 ± 0.5 | 0.33 |
| 1 June 2016 | 183.2 ± 7.3 | 9.1 ± 1.4 | 8.6 ± 0.5 | 0.13 |
| 24 June 2016 | 227.8 ± 8.7 | 14.4 ± 1.7 | 16.8 ± 0.8 | 0.01 |
| 4 Aug 2016 | 272.4 ± 9.7 | 22.5 ± 1.8 | 23.7 ± 1.6 | 0.65 |
| 16 Sept 2016 | 301.9 ± 10.9 | 26.7 ± 1.0 | 26.2 ± 1.4 | 0.19 |
| 7 Oct 2016 | 296.7 ± 8.6 | 23.3 ± 1.4 | 23.7 ± 1.3 | 0.42 |
| 28 Oct 2016 | 251.1 ± 7.3 | 25.6 ± 0.8 | 25.3 ± 1.1 | 0.29 |
| 1 Dec 2016 | 233.7 ± 5.9 | 21.8 ± 1.0 | 22.0 ± 1.1 | 0.28 |
| 30 Dec 2016 | 178.5 ± 6.1 | 22.3 ± 1.3 | 22.3 ± 1.3 | 0.59 |
| 29 Jan 2017 | 172.5 ± 5.6 | 18.2 ± 1.5 | 18.9 ± 0.9 | 0.20 |
| 2 Mar 2017 | 186.2 ± 8.5 | 13.7 ± 1.2 | 14.1 ± 0.8 | 0.29 |
| 29 Mar 2017 | 168.9 ± 6.2 | 17.6 ± 1.6 | 17.3 ± 0.9 | 0.52 |
| 3 May 2017 | 157.7 ± 6.9 | 14.7 ± 2.2 | 20.0 ± 1.3 | 0.59 |
| 15 June 2017 | 199.1 ± 8.4 | 10.6 ± 1.3 | 12.3 ± 0.7 | 0.75 |
| 21 July 2017 | 194.4 ± 7.1 | 9.0 ± 0.5 | 9.1 ± 0.4 | 0.13 |
| 16 Aug 2017 | 240.4 ± 6.6 | 12.7 ± 1.1 | 13.2 ± 0.6 | 0.16 |
| 14 Sept 2017 | 274.3 ± 6.6 | 18.4 ± 1.0 | 18.1 ± 0.7 | 0.52 |
| 27 Oct 2017 | 263.0 ± 6.1 | 21.4 ± 1.5 | 21.1 ± 1.0 | 0.59 |

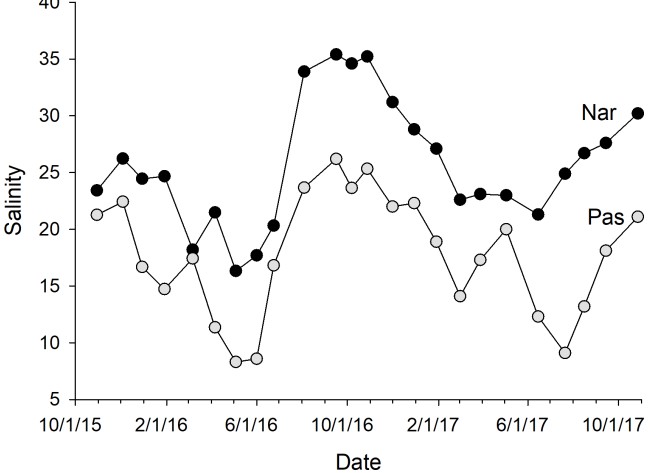

**Figure 4** **Plot of mean whole-marsh calculated salinity versus day of sampling for the NAR and PAS study sites.** The date of the initial sample event October 30, 2015 was designated as day 1. Sample minima at days 188 and 631 corresponded to the dates May 4, 2016 and July 21, 2017. Sample maxima at days 323 and 729 corresponded to the dates September 16, 2016 and October 27, 2017.

**Table 3** **Best predictive models incorporating the effect of cumulative rainfall amounts and tide state at time of sampling on calculated salinity values during 24 sample events at the (a) southern (NAR) and (b) northern (PAS) study sites.** Models best supported by the data, or those having $\Delta AIC_c$ values between 0.00 and 2.00, are listed.

**(a.)**

| NAR model[a] | $R^2$ | $AIC_c$ | $\Delta AIC_c$[b] |
|---|---|---|---|
| $48.77 - 0.831(6\ \text{MON}) - 3.193(\text{TIDE})$ | 0.55 | 69.12 | 0.00 |
| $44.22 - 0.781(6\ \text{MON})$ | 0.48 | 70.13 | 1.01 |
| $47.54 - 0.873(6\ \text{MON}) - 3.124(\text{TIDE}) + 0.531(1\ \text{MON})$ | 0.57 | 70.78 | 1.66 |

**(b.)**

| PAS model[a] | $R^2$ | $AIC_c$ | $\Delta AIC_c$[b] |
|---|---|---|---|
| $37.41 - 0.873(6\ \text{MON}) - 3.124(\text{TIDE}) + 0.531(1\ \text{MON})$ | 0.80 | 50.55 | 0.00 |

Notes.

[a] 1 MON, cumulative rainfall 30 days prior to sample event; 6 MON, cumulative rainfall 180 days prior to sample event; TIDE, tide state.

[b] $\Delta AIC_c = AIC_{ci} - AIC_{cmin}$.

**Table 4** **Relative importance of rainfall and tide parameters in regression models explaining calculated salinity values during 24 sample events at the southern (NAR) and northern (PAS) study sites.**

| Parameter | NAR relative importance | PAS relative importance |
|---|---|---|
| 24 HR | 0.195 | 0.188 |
| 1 MON | 0.327 | 0.999 |
| 6 MON | 1.000 | 1.000 |
| TIDE | 0.596 | 0.966 |

**Table 5** **Mean bulk density and percent moisture in soil samples to 25 cm depth collected in high, mid, and low marsh locations at the southern (NAR) and northern (PAS) study sites.**

| Site | Location | Bulk density (g cm$^{-3}$) | Percent moisture (%) |
|---|---|---|---|
| NAR | High marsh | $0.31 \pm 0.04$ | $71.3 \pm 3.3$ |
| NAR | Mid marsh | $0.19 \pm 0.01$ | $84.3 \pm 1.4$ |
| NAR | Low marsh | $0.34 \pm 0.29$ | $69.0 \pm 16.8$ |
| PAS | High marsh | $0.24 \pm 0.05$ | $77.1 \pm 0.6$ |
| PAS | Mid marsh | $0.17 \pm 0.02$ | $84.2 \pm 1.4$ |
| PAS | Low marsh | $0.16 \pm 0.03$ | $83.2 \pm 2.4$ |

consistently higher than that at PAS, which is located approximately 35 km to the north near the head of the estuary. Mean surface seawater salinity at a long-term water quality sample site in Narragansett Bay, located approximately 1 km north of PAS, averaged 25.1 $\pm$ 0.8 ppt, while a site approximately 4 km north of NAR averaged 31.5 $\pm$ 0.2 ppt (R McKinney, 2018, unpublished data). These values should approximate the salinity of the seawater inundating each marsh during flood tides. Salinity of freshwater sources would likely vary somewhat both spatially and temporally, but most likely had salinities less than 5 ppt (*Dodds, 2002*). Nothing is known of the relative contribution of each salinity end-member to porewater salinity at each site, still it is likely that the lower seawater salinity near PAS contributed to the lower mean calculated salinities we observed.

**Table 6** Least squares regression statistics for the relationship between calculated salinity and elevation for a sub-set of 12 sample events corresponding to calculated salinity maxima and minima over the course of the study at the (a) southern (NAR) and (b) northern (PAS) study sites.

**(a.)**

*NAR Minima*

| Sample date | Slope | $R^2$ | Degrees of freedom | $p$ |
|---|---|---|---|---|
| 4/6/2016 | −8.42 | 0.35 | 44 | <0.001 |
| 5/4/2016 | −25.47 | 0.54 | 52 | <0.001 |
| 6/1/2016 | −38.62 | 0.42 | 34 | <0.001 |
| 6/15/2017 | −21.72 | 0.55 | 58 | <0.001 |
| 7/21/2017 | −36.09 | 0.59 | 60 | <0.001 |
| 8/16/2017 | −24.99 | 0.63 | 61 | <0.001 |

*NAR Maxima*

| Sample Date | Slope | $R^2$ | Degrees of freedom | $p$ |
|---|---|---|---|---|
| 9/16/2016 | −11.90 | 0.61 | 47 | <0.001 |
| 10/7/2016 | −15.56 | 0.57 | 52 | <0.001 |
| 10/28/2016 | −9.75 | 0.33 | 54 | <0.001 |
| 9/14/2017 | −27.62 | 0.53 | 76 | <0.001 |
| 10/27/2017 | −18.05 | 0.57 | 54 | <0.001 |
| 11/21/2017 | −24.68 | 0.55 | 61 | <0.001 |

**(b.)**

*PAS Minima*

| Sample date | Slope | $R^2$ | Degrees of freedom | $p$ |
|---|---|---|---|---|
| 4/6/2016 | 0.71 | 0.01 | 25 | 0.643 |
| 5/4/2016 | −0.34 | 0.01 | 30 | 0.556 |
| 6/1/2016 | −0.67 | 0.07 | 30 | 0.149 |
| 6/15/2017 | −4.10 | 0.37 | 36 | <0.001 |
| 7/21/2017 | 0.03 | 0.01 | 29 | 0.897 |
| 8/16/2017 | −1.11 | 0.20 | 33 | 0.007 |

*PAS Maxima*

| Sample date | Slope | $R^2$ | Degrees of freedom | $p$ |
|---|---|---|---|---|
| 9/16/2016 | −0.25 | 0.00 | 25 | 0.746 |
| 10/7/2016 | −4.54 | 0.34 | 30 | <0.001 |
| 10/28/2016 | 0.89 | 0.07 | 31 | 0.124 |
| 9/14/2017 | −1.51 | 0.12 | 39 | 0.024 |
| 10/27/2017 | −0.51 | 0.01 | 33 | 0.601 |
| 11/21/2017 | −0.11 | 0.00 | 35 | 0.907 |

Mean calculated salinities for the marshes showed maxima roughly corresponding to late summer, when plant biomass is high and evapotranspiration is assumed to be at its peak, and minima in early to mid-spring when evapotranspiration is low and snow melt and rainfall could lead to increased freshwater input to the marshes. Several studies have suggested a conceptual model of factors influencing near-surface tidal marsh porewater salinity, lower salinity freshwater inputs arising from groundwater flow under the marsh and surface water inputs interacting with periodic inputs of higher salinity seawater delivered

during semi-diurnal flood tides (*Barry, Barry & Parlange, 1996*; *Li & Jiao, 2003*; *Parlange et al., 1984*). Variation in the position of the water table both spatially and temporally will determine soil saturation patterns and will influence observed soil salinities across the marsh surface (*Montalto, Parlange & Steenhuis, 2007*). Results of multiple linear regression models of cumulative regional rainfall, a driver of groundwater and surface water inputs, and tide state versus our observed mean salinities in the marsh lend some support to this model at our sites, with longer-term cumulative rainfall showing a greater relative importance in our models than shorter-term precipitation, particularly at the PAS site. Longer-term cumulative rainfall patterns may be more indicative of the magnitude of groundwater flow to coastal marshes if groundwater flow in the watershed is relatively slow, say on the order of $0.002$ m day$^{-1}$ as predicted in soils with hydraulic conductivity of $0.01$ m day$^{-1}$ (*Heath, 1983*). However, many other factors not measured or accounted for in our study, including the timing and magnitude of evapotranspiration, groundwater flow patterns under a marsh, marsh topography, mean temperature, and variability in tidal inundation patterns will interact to influence soil saturation and observed patterns of soil salinity across a marsh.

In soils with similar clay and organic matter content, EC$_a$ values will respond to changes in soil composition, bulk density, moisture content, and soil salinity (*Corwin & Lesch, 2005*). Previous studies have suggested EC$_a$ could be a reliable means to rapidly assess soil salinity, particularly in hydric soils (*Sheets, Taylor & Hendrickx, 1994*; *Hanson & Kaita, 1997*). In homogenous, uniformly saturated salt marsh soils it may be reasonable to assume that EC$_a$ may accurately reflect changes in soil salinity. However, regression statistics of the equations used to generate our calculated salinity values, for example the variable correlation coefficient and slope values observed, could be an indication that other soil parameters may be influencing EC$_a$ values at our sites. Soils at our sites were consistently at or around 70% moisture, suggesting uniformly saturated soils that would satisfy this assumption of the technique. We did see some intra-marsh differences in soil bulk density at PAS that may have contributed somewhat to variability in EC$_a$ values. It may also be possible that our samples may have reflected spatial variation in soil composition at the sites: if different regions of the marsh differed in soil composition, combining calibration data across these regions may increase observed variability. Another possible explanation could be non-homogeneous presence of conductive clay minerals or iron sulfate in the soils, both of which may directly impact EC$_a$ values (*Laforet, 2011*).

Variability in regression statistics can also be the result of spatial variability in porewater salinity values, from vagaries in water table levels or groundwater flow at out sites. For example, in our study EC$_a$ values reflected soil characteristics to 0.5 m below the marsh surface, while porewater salinities used in the calibration equations were measured at a depth of 25 cm. Spatial variability in soil porewater salinity either above or below our porewater sample depth would be reflected in EC$_a$ values, but not necessarily in our measured porewater salinity values. Mean plant root biomass at our sites is assumed to be around 0.4 m below the surface and may impact deeper porewater dynamics that could would affect EC$_a$ values but not be reflected in our porewater salinities. During seasonal extremes in salinity this could significantly influence interpretation or misinterpret actual

conditions in the rhizosphere that affect salinity-driven plant zonation patterns. Addition of a second, deeper porewater salinity sampling point may help to resolve this potential confounding factor. Differences in soil saturation may also have influenced our measured $EC_a$ values, although to what extent is not clear. In a model of water table dynamics and groundwater movement in a tidal marsh, *Ursino, Silvestri & Marani (2004)* found that a zone of unsaturated, aerated soil could form in a marsh in areas away from the hydraulic influence of tidal creeks, and that this aerated zone could migrate toward the inner part of the marsh over time. They also found that evapotranspiration can result in the formation of an unsaturated aerated layer trapped underneath saturated surface soil, particularly in areas away from the influence of tidal creek hydrology (*Ursino, Silvestri & Marani, 2004*). Either of these phenomena could impact $EC_a$ values while conceivably not impacting measured porewater salinity, and hence may contribute to the variability in calibration statistics.

Correlations of calculated salinity with marsh elevation supported our qualitative assessment of intra-marsh salinity variation shown by the contour plots. Calculated salinity at NAR significantly correlated with elevation over all the examined sample events, reinforcing observed patterns of higher soil salinity near the creek edge and lower salinity towards the upland border. Previous studies have documented increases with soil elevation, reaching a maximum just above mean high sea level and decreasing towards the upland edge of this marsh (*Mahal & Park, 1976*; *Adam, 1990*). These observations could be attributed to progressively less frequent flooding of the marsh and the associated reduced salt input at higher elevations near the marsh upland border (*Adam, 1990*). At very high soil elevations, above MHSL, soil water salinity tends to decrease due to. At PAS, the lack of significant correlation may have resulted from the more homogenous salinity patterns observed across the marsh surface, or may have reflected the predominance of groundwater or surface freshwater inputs at the site.

## CONCLUSIONS

Results of our study suggest that despite variability in calibration coefficients, $EC_a$ values reflect longer-term changes in porewater salinity at a single marsh. Therefore, $EC_a$ values show promise in tracking spatial patterns of soil salinity over time at a given site, which could aid in identifying changes in marsh biogeochemistry that could ultimately impact plant zonation. This is particularly true under the assumption that ECa values are a dependable proxy for direct porewater sampling once calibrated with actual field data: the relative ease of this technique makes mapping large or repeated spatial areas with EMI far more efficient than traditional approaches. For example, $EC_a$ surveys of a marsh may aid in identifying areas of irregular seawater or freshwater infiltration and help increase our understanding of marsh hydrology at a given site. Several studies are underway in northeast US salt marshes to document shifts in high and low marsh plant communities, in the context of increased flooding from sea level rise. Fine scale mapping of salinity using EMI may aid in determining salinity patterns that will drive these shifts before the plant species migrate. In this way, ECa mapping may aid in restoration planning and monitoring, especially of low-lying coastal salt marshes vulnerable to sea level rise. However, our results

also suggest that inter-marsh comparisons of $EC_a$ values and calculated salinities should be interpreted with caution: to accurately compare values, soil composition will either need to be similar, or between marsh differences adequately characterized and considered during the calibration process.

## ACKNOWLEDGEMENTS

We thank Nicole Gutierrez and Katelyn Szura for assistance with initial sample protocol development, and Jara Botelho, Jess Janiec, and Tia Mitchell for assistance with data collection. Cathy Wigand, Sandi Robinson, Steve Shivers and Suzy Ayvazian provided helpful input on an earlier version of this manuscript. The views expressed in this paper are those of the authors and do not necessarily reflect the views or policies of the US Environmental Protection Agency. This contribution is identified by Tracking Number ORD-028876 of the Atlantic Ecology Division, Office of Research and Development, National Health and Environmental Effects Research Laboratory. Mention of trade names, products, or services does not convey, and should not be interpreted as conveying, official EPA approval, endorsement, or recommendation.

### Funding

Funding for this work was provided by the US Environmental Protection Agency. The funders had no role in study design, data collection and analysis, decision to publish, or preparation of the manuscript.

### Grant Disclosures

The following grant information was disclosed by the authors:
US Environmental Protection Agency.

### Competing Interests

Michael Charpentier is employed by General Dynamics Information Technology.

### Author Contributions

- Richard McKinney conceived and designed the experiments, performed the experiments, analyzed the data, prepared figures and/or tables, authored or reviewed drafts of the paper, approved the final draft.
- Alana Hanson conceived and designed the experiments, performed the experiments, authored or reviewed drafts of the paper, approved the final draft.
- Roxanne Johnson conceived and designed the experiments, performed the experiments, prepared figures and/or tables, authored or reviewed drafts of the paper, approved the final draft.
- Michael Charpentier performed the experiments, analyzed the data, contributed reagents/materials/analysis tools, prepared figures and/or tables, authored or reviewed drafts of the paper, approved the final draft.

### Field Study Permissions

The following information was supplied relating to field study approvals (i.e., approving body and any reference numbers):

Permission for non-invasive field study was provided by RI Department of Environmental Management, under collection permits #2015-31-F–2018-31-F.

### Data Availability

Raw measurements are available in the Supplemental Files.

### Supplemental Information

Supplemental information for this article can be found online at http://dx.doi.org/10.7717/peerj.8074#supplemental-information.

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
