# Peer review of "Seasonal variation in apparent conductivity and soil salinity at two Narragansett Bay, RI salt marshes"

_PeerJ, doi:10.7717/peerj.8074_

## Round 0.1 · original submission · Major Revisions

The reviewers have provided very useful comments to improve the manuscript. Please consider these constructive comments and I look forward to your revised manuscript.

Reviewer 1 ·

Basic reporting

No comment.

Experimental design

The methods should be improved or revised prior to publication to improve the strength of the manuscript. First, I had questions about the spatial interpolation method, that should be addressed in the context of the manuscript revision.

First, quantification of uncertainty in the salinity calculations needs to be done explicitly.

Second, how was it decided to use this method for interpolation mapping? Where any validation techniques used to arrive at this method, such as leaving out a sub-set of data points, and estimating prediction error? Did you walk specific transects when you did the surveys? If not, do you think that you produced different maps because different paths were taken during different seasons? Did you do any analysis of how maps might vary based on different paths taken? How closely spaced where your measurements and what role do you think that spatial autocorrelation plays in your analysis?

Third, why were RTK-GPS measures made of elevations if LiDAR, with many thousands of points, and much more elevation detail, is available? How do your generated DEMs compare with the bare earth LiDAR?

Fourth, are the cumulative rainfall values correlated? Does this influence your data analysis? I would recommend trying out different synthetic lag-terms as well to see if some fit your data better than others.

I have added specific line comments below, as well.

Line 143 – At how many points were sipper readings taken per survey?

Line 163 – Again, is there any evidence that this interpolation method is the most appropriate and accurate for mapping elevation? And why not use LiDAR?

Line 168 – How did you adjust NAVD88 to sea level ?

Line 196- Explain how spatially averaged salinity was calculated.

Line 196-7 – Also was there consistent variability by season? That could be examined for using something like GLM or linear mixed effects models?

Validity of the findings

The discussion of findings could be improved by more reference and discussion of primary literature on spatial structure of salinity gradients in coastal wetlands.

Additional comments

This manuscript focuses on the use of electro-magnetic conduction surveys to identify spatial and temporal patterns in coastal marsh salinity. This can make a valuable contribution to the literature. Below I have identified locations where the manuscript prose can be improved to increase clarity.

Line 72 – The phrasing in this sentence is somewhat awkward and should be revised. Tidal wetlands have a continuously saturated zone, and so it is not that porewater is “absent;” it can just be difficult to sample if the saturated zone is deep, or if there are clay or fine sediment layers with low hydraulic conductivity.

Line 90- I know that EMI is defined in the abstract, but it would be helpful to define it at first use in the manuscript as well.

Line 105 – I suggest replacing the second “and” with a comma.

Line 122/123 replace spp with spp.

Line 122- Likely your “Scirpus” species have been renamed to genera such as Schoenoplectus and Bolboschenus.


Line 183 – What models? Models haven’t been discussed yet.

Line 191-194- This is unclear and needs to be better explained. What is 1.0 to the proportions?

Line 212 – Producing biplots is not a quantitative assessment of association I would recommend removing completely the “visual comparison of maps.”

Line 220-1 – This sentence could be more clearly written.

Line 222-4 - Can this be written more simply? Such as, “salinity was high at the seaward edge and low at the terrestrial border across the spring to fall growing season”

Results section should present results of calibration of porewater salinity vs. conductivity and prediction errors (ie, not just R2, but actual prediction errors). Bulk density was measured – how does that play into the calibration?

Line 230 – “sampling” not “sample” ? Also, these values seem overly precise given unspecified prediction errors that are likely quite large (r2~0.5), e.g., salinity +/- 10‰.

Line 239 – Are you referring to mean salinity for the whole marsh?

Line 266-267 – This sentence could be more specific. What were the results, beyond salinity varied spatially and temporally?

Line 268 – Why was this salinity pattern expected?

Line 269-272 – This section is results.

Line 274- I’m not sure that a lower water table in the marsh would be caused by reduced precipitation. Isn’t the source of most of the shallow groundwater the tides? Do you have evidence for lower water tables in marshes during droughts? Maybe revise “lower water table” to “reduced groundwater recharge from uplands.”

Discussion – Different coastal marsh salinity patterns are found in different places based on seasonality in precipitation and/or tidal flooding (e.g., some marshes are flooded more by wind events than tides). The discussion should be revised to identify these differences. As it stands now, statements such as “Seawater influence has been shown to diminish as distance from tidal creeks increases (Hemmond and Fifield 1982)” are overly centric on northeastern marshes. While these marshes are well studied, the discussion should be drawn on more from the international literature to show how these surveys both confirm and refute past studies of northeastern marshes, and how this approach could be valuable in e.g., Mediterranean marshes that are often hypersaline at the marsh upland border.

Do these marshes have similar hydraulic conductivity values? Are the tides the same? Wouldn’t these things help explain differences in spatial patterns of salinity?

Line 284 – It doesn’t follow to me that high freshwater discharge to the marsh would translate into homogenous salinity patterns.

Line 288-291 – The prose could be simplified here.

Line 299 – Why would freshwater have a salinity of 5ppt? Isn’t that by definition not freshwater?

Line 319 – Can you be more specific than “relatively slow” ? Days? Months? Years?

Line 362 – Sentence is unclear.

Line 368 – The implications here seem not specific enough. Refer more specifically to spatial patterns in salinity and plant zonation.

Line 372 – Vulnerable to what? Erosion? Salt water intrusion?

Table 1 – How many points were sampled for porewater conductivity? Normally, dates should be in international format, e.g., 30 Oct 2015, as most countries use the day-month-year formate.

Figure 1 – “Salt marsh study sites” not “Salt marshes study sites”

Figure 2 & 3 – Sampling points should be demarcated. Also, it would be easier to interpret this maps if they had the same legend (e.g., yellow represented the same thing on the different plots). If they are - I apologize – it is hard to tell. Perhaps just have one legend that shows all the possible salinity levels?

Figure 4 – x axis should be “day of year-” Also I suggest making the figure more professional, rather than using excel default style. I suggest the authors look at figures published in journals, especially PeerJ for examples of publication quality figures.

Reviewer 2 ·

Basic reporting

No comment

Experimental design

No Comment

Validity of the findings

No Comment

Additional comments

The authors have presented an interesting study that follows up prior work using electromagnetic induction (EMI) and apparent conductivity ECa measures as a valid/dependable proxy for pore water salinity measures in marshes, clearly demonstrating EMI’s value in tracking interannual changes in soil salinity patterns in a new and highly efficient manner.

I think they have done a fine job in designing, implementing and interpreting their study for the most part. That said, I have some concerns that I’d want the authors to consider and explain, as well as some minor comments and suggestions they could address to help make the study more valuable to readers and/or users of EMI technology in the future.

One such concern is why they did not interpret results by major habitat type (high vs low marsh for example) and rather treated the marsh as a whole/single unit. My sense is that some interesting patterns were lost by not considering how the results shake out for low vs high marsh areas, esp since different plants, different rates of water uptake, different root depth and biomass/bulk density, and often different soil types (greater percentages of silt/clay, for example that can affect ECa). This is a fact the Authors note in Lines 337-338, but more could be done to account for potential influence of clay minerals/iron sulfate soils. While they did list the dominant plant species (Lines 119-129), there may be more analysis to be done here. Keep in mind most researchers in New England are documenting shifts in high/low marsh plant communities, where low marshes are drowning and high marshes are converting to short form S. alterniflora or worse… EMI used as detailed in this manuscript may help show the salinity patterns that will drive these shifts BEFORE the plants are forced to react and migrate. THAT is very interesting and underplayed in the ms as presently written…

Lines 138-139: Were pore water samples (via sipper) gathered at each monthly sampling event or were representative calibration samples taken at some other time of year and used to correct/convert ECa to salinity at each monthly sampling event? Obtaining pore water in Dec-Mar must have been a real challenge!! Must be warmer than I thought in Narragansetts Bay in mid-winter! Table 1 suggests actual pore water sample collected each month, correct? Please clarify.

Lines 143-145: The Authors note their pore water sampling depth of 0.25 m, which is rather shallow. Depending on the plant community, that mean depth of live roots may be closer to 0.40 m and often there is a marked difference in pore water salinity near the surface (< 0.10m) and deeper depth (~ 0.06m). My concern is that using the 0.25m as the calibration value against ECa via EMI is mismatched. EMI integrates the entire profile, with the sipper is a fixed point. During seasonal extremes (min/max salinity) this could significantly influence interpretation or misinterpret actual conditions in the rhizosphere that affect plant zonation patterns. And Typha (present at the upper edge of the marsh can produce significant root/rhizomes well below 0.25m…, for example). Sampling at two depths might have resolved this potential confounding factor. Since they can’t go back and take that second depth, they should do more to acknowledge the potential flaw (in addition to text they already provided in Lines 341-345).

Lines 267-269: I have often found a bimodal pattern of salinity along a gradient of creek edge to upland, with the highest salinity often midway along that gradient of well-drained creek edges and freshwater collecting upland edges. High marsh can often collect and retain salty water as freshwater evaporates out… However, in smaller marshes, the pattern may be more as the Authors describe. I guess I just caution them against saying “expected pattern” without qualifying why its “expected”.

Lines 366-368: This line underscores my concern that vegetation data is lacking in this manuscript. At the very least, consider results by high vs low marsh habitat.

Lines 365-375: In my opinion, the Authors miss a good opportunity to highlight how efficient EMI can be for this type of sampling. If we are to believe that ECa is a dependable proxy for direct pore water sampling once calibrated with actual field data, then we should be equally convinced that mapping large or repeated spatial areas with EMI is far more efficient than traditional approaches – its wouldn't be impossible to do this with sippers, but it would have taken 10-100x more person hours/effort! EMI makes this type of study practical/possible.

Table 1: So, the Authors were able to extract a field pore water sample in Dec, Jan, Feb, etc.? Impressive!

Table 4: Since the Authors indicate bulk density samples were obtained from low vs high marsh habitats, I’m hoping they can sort out which ECa and pore water samples came from which habitat and will consider analysis of results through this additional lens. This is PARTICULARTLY important as New England is watching high marsh convert to Low marsh due to SLR and prolonged flooding. EMI data will track nicely (if not be predictive of) vegetation pattern shifts… I STRONGELY ENCOURAGE THE AUTHORS TO RECAST RESULTS/DISCUSSION IN THIS TIMELY PERSPECTIVE.

Figures 2-3: Add a high/low marsh line to maps to help direct readers to potential patterns in each key habitat.

---

## Round 0.2 · accepted · Accept

Your thoughtful responses to the reviewer’s concerns are greatly appreciated. This is an excellent contribution. Thank you!

Reviewer 2 ·

Basic reporting

See recommendation below.

Experimental design

See recommendation below.

Validity of the findings

See recommendation below.

Additional comments

After careful review of the revised manuscript, I believe the authors have made all necessary revisions (or provided acceptable explanations) to warrant publication as is.